# Comparison of Contributors to Mortality Differences in SLE Patients with Different Initial Disease Activity: A Larger Multicenter Cohort Study

**DOI:** 10.3390/jcm12031061

**Published:** 2023-01-30

**Authors:** Ziyi Jin, Zheng Chen, Wenyou Pan, Lin Liu, Min Wu, Huaixia Hu, Xiang Ding, Hua Wei, Yaohong Zou, Xian Qian, Meimei Wang, Jian Wu, Juan Tao, Jun Tan, Zhanyun Da, Miaojia Zhang, Jing Li, Xuebing Feng, Lingyun Sun

**Affiliations:** 1Department of Rheumatology and Immunology, The Affiliated Drum Tower Hospital of Nanjing University Medical School, Nanjing 210008, China; 2Department of Rheumatology, Huai’an First People’s Hospital, Huai’an 223001, China; 3Department of Rheumatology, Xuzhou Central Hospital, Xuzhou 221008, China; 4Department of Rheumatology, The Third Affiliated Hospital of Soochow University, Changzhou 213003, China; 5Department of Rheumatology, Lianyungang Second People’s Hospital, Lianyungang 222000, China; 6Department of Rheumatology, Lianyungang First People’s Hospital, Lianyungang 222002, China; 7Department of Rheumatology, Northern Jiangsu People’s Hospital, Yangzhou 225007, China; 8Department of Rheumatology, Wuxi People’s Hospital, Wuxi 214028, China; 9Department of Rheumatology, Jiangsu Province Hospital of TCM, Nanjing 210004, China; 10Department of Rheumatology, Southeast University Zhongda Hospital, Nanjing 210009, China; 11Department of Rheumatology, The First Affiliated Hospital of Soochow University, Suzhou 215005, China; 12Department of Rheumatology, Wuxi TCM Hospital, Wuxi 214177, China; 13Department of Rheumatology, Zhenjiang First People’s Hospital, Zhenjiang 212050, China; 14Department of Rheumatology, Affiliated Hospital of Nantong University, Nantong 226001, China; 15Department of Rheumatology, Jiangsu Province Hospital, Nanjing 210029, China; 16Department of Rheumatology, Affiliated Hospital of Jiangsu University, Zhenjiang 212050, China

**Keywords:** SLE, mortality, attributable risk, relative risk, disease activity

## Abstract

To explore the etiology of risk factors and quantify the mortality differences in systemic lupus erythematosus (SLE) patients with different initial disease activity. The Jiangsu Lupus database was established by collecting medical records from first-hospitalized SLE patients during 1999–2009 from 26 centers in Jiangsu province, China, and their survival status every five years. The initial SLEDAI scores [high (>12) vs. low–moderate (≤12)] differences in mortality attributable to risk factors were quantified using population attributable fraction (PAF), relative attributable risk (RAR) and adjusted relative risk (ARR). Among 2446 SLE patients, 83 and 176 deaths were observed in the low–moderate and high activity groups, with mortality rates of 7.7 and 14.0 per 1000 person years, respectively. Anemia was the leading contributor to mortality, with PAFs of 40.4 and 37.5 in the low–moderate and high activity groups, respectively, and explained 23.2% of the mortality differences with an ARR of 1.66 between the two groups. Cardiopulmonary involvement caused the highest PAFs in the low–moderate (20.5%) and high activity (13.6%) groups, explaining 18.3% of the mortality differences. The combination of anemia and cardiopulmonary involvement had the highest RAR, causing 39.8% of the mortality differences (ARR = 1.52) between the two groups. In addition, hypoalbuminemia and a decrease in the creatinine clearance rate accounted for 20–30% of deaths and explained 10–20% of the mortality differences between the two groups, while antimalarial drug nonuse accounted for about 35% of deaths and explained 3.6% of the mortality differences. Anemia, cardiopulmonary involvement and hypoalbuminemia may cause substantial mortality differences across disease activity states, suggesting additional strategies beyond disease activity assessment to monitor SLE outcomes.

## 1. Introduction

Systemic lupus erythematosus (SLE) is a chronic multisystem inflammatory autoimmune disease and affects multiple organs damage that may be fatal. Advances in therapies have substantially improved the survival rate of SLE patients, especially between the 1950s and mid-1990s but then kept steady in the last two decades [1]. Through the 2000s, the global 5-year, 10-year and 15-year survival rates of adult SLE patients were 0.95, 0.89 and 0.82 in high-income countries, respectively, and further slightly reduced in low/middle-income countries. In addition, when compared with the general population, the standardized mortality ratio in SLE patients persists to be high, with 2–5 in the world [2,3]. Thus, some patients with SLE still have high risk of deaths even under the same appropriate treatments as others.

The survival rate of SLE is diverse and attributed to many factors. Lots of epidemiological studies have shown that SLE survival rates are affected by race, socio-demographic, genetic and clinical factors, of which disease activity, infection, renal failure and cardiovascular (CVD) complications have been well established as the main causes of SLE deaths [4]. Moreover, high disease activity was positively associated with infection, organ damage, CVD risk and other risk factors of death, leading to a much higher risk of death among patients with SLE [5,6,7,8,9]. Although the risk factors of SLE deaths have been identified frequently, little attention has been paid to quantifying the risk attributable to specific factors and what fraction of the differences in SLE mortality in different disease activity subgroups are due to differences in risk factor exposure. To answer the above questions, the three measures of population attributable fraction (PAF), relative attributable risk (RAR) and adjusted relative risk (ARR), which have been widely used in other diseases, were needed [10,11,12]. PAF is a widely used index to measure the proportion of deaths that is attributed to one or more specific causes and could be prevented if those causes were eliminated. In addition, the RAR was extended from the PAF to measure the proportion of observed events excess in the high-risk group that would vanish if it had the same distribution of risk factors as the low-risk group. The ARR is the ratio of rates in the high-risk group to the low-risk group that would be observed where the distribution of risk factors was the same.

In this study, we separately examined the associated risk factors of SLE deaths in low–moderate and high activity groups classified by initial disease activity, evaluated the role of major risk factors on the PAF of SLE deaths in the two groups and investigated what proportion of the mortality gradient between the two groups could be explained by differences in the distribution of the risk factors.

## 2. Methods

### 2.1. Study Design and Patients

This study has been described in recent published studies [13,14]. In brief, a large-scale multicenter retrospective cohort study was carried out among 26 centers in Jiangsu province, China, by the Jiangsu Lupus Collaborative Group in 2010. Participants were first-hospitalized SLE patients with complete medical records between 1 January 1999 and 31 December 2009. The diagnostic criteria met at least four classification criteria of SLE revised and updated by the ACR [15]. So far, follow-ups have been conducted three times to collect medical records and survival status of participants in 2010, 2015 and 2021, respectively. This study was approved by the Institutional Review Board of Nanjing Drum Tower Hospital.

### 2.2. Data Collection and Definition

We built a website (http://sys.91sqs.net/sle/Index/index.html, accessed on 28 January 2023) to gather and manage the longitudinal data from 26 multicenter hospitals. Information of patients with SLE including demographic data, clinical features, diseases history, organ involvement, laboratory tests and medication use were extracted from inpatient medical record information system.

The SLE disease activity index 2000 (SLEDAI) score was used to assess disease activity, and a score ≤6 was defined as no and mild activity, 7–12 was moderate median activity and >12 was high activity [16]. Consequently, we divided SLE patients into two groups: high activity group (SLEDAI > 12) and low–moderate activity group. Organ damage was measured using SLICC/ACR damage index [17], and the manifestations in specific organ involvement were listed in Appendix A. Cardiopulmonary involvement mainly included serositis, interstitial lung disease, myocarditis, pulmonary arterial hypertension and a few other manifestations such as pulmonary hemorrhage/vasculitis, valvular dysfunction, arrhythmia and cardiac failure. Diagnosis time was calculated from the time at disease onset to the time of diagnosis. Survival time was calculated from the date of diagnosis of SLE to the date of death or the date of the last follow-up for censored survivors. The laboratory variables were dichotomized into normal and abnormal groups, and the following categories were defined as abnormal groups: low red blood cell count (3.5 × 10^12^/L for female and 4 × 10^12^/L for male), low hemoglobin (110 g/L for female and 120 g/L for male), abnormal white blood cell count (<4 × 10^9^/L or >10 × 10^9^/L), hypoalbuminemia (<35 g/L), decrease in creatinine clearance rate [<70 mL/min, calculated by (140-age) × weight × (0.85, if female)/(72 × serum creatinine)], anti-dsDNA positive (immunoblotting or >100 using enzyme immunoassays), antinuclear antibody positive (>1:40 using immunofluorescence method), anti-Sm positive (immunoblotting), abnormal C3 complement (<0.8 g/L) and abnormal C4 complement (<0.2 g/L). Anemia included either low red blood cell count or low hemoglobin. Immunosuppressive drugs included antimalarial drugs, cyclophosphamide, leflunomide, azathioprine, methotrexate, mycophenolate mofetil, tripterygium, ciclosporin and tacrolimus. According to the proportion, we analyzed antimalarial drugs and cyclophosphamide separately and combined others as other immunosuppressives. Biologic drugs, such as rituximab and infliximab, were also collected but not analyzed due to the very small sample size.

### 2.3. Statistical Analysis

Ages and SLEDAI scores in a non-normal distribution were described in median (interquartile range (IQR)) and compared using the Mann–Whitney U test between low–moderate and high activity groups. The distribution of categorical variables was compared using the χ^2^ test. The survival rates for two groups were depicted using Kaplan–Meier survival curves and compared using log-rank tests. Hazard ratios (HRs) and corresponding 95% confidence intervals (CIs) were used to quantify the magnitude of associations using Cox proportional hazard regression models. Moreover, the restricted cubic spline model was applied to explore possible nonlinear dose–response relationship between the SLEDAI score at admission and mortality of SLE. Multivariate analyses were carried out when considering the following confounders: gender (male = 1, female = 0), age (continuous), SLEDAI score at admission (continuous), comorbidities (yes = 1, no = 0), glucocorticoid treatment on admission (yes = 1, no = 0) and immunosuppressive treatment on admission (yes = 1, no = 0).

The adjusted PAF was calculated to estimate the proportion of deaths that would not have occurred if the risk factors of interest had been absent or at baseline levels. The formula was described by Bruzzi et al. as follows [18]:$$\mathrm{PAF}=1-\sum_{j=1}^{J}\pi_{j}\psi_{j}^{-1}$$
where *π_j_* denotes the proportion of deaths at exposure level *j*, and *ψ_j_* denotes the exposure-specific rate ratios that can be approximately equal to HR modelled by multivariate analyses in Cox proportional hazard models. The 95% CI of PAF was obtained using the method based on the Bonferonni inequality described by Natarajan et al. [19]. The joint PAF for a combination of some selected risk factors were estimated to observe the proportion of deaths that can be attributed to any of the risk factors or their combinations [20].

The relative attributable risk (RAR) and the adjusted relative risk (ARR) were further calculated to measure how much of different mortality rates were due to the difference in patterns of risk factors, as suggested by Lele and Whittemore [21].
(1)$$\mathrm{RAR}=\frac{1-\left(1-{\mathrm{PAF}}_{H}\right)(\sum_{j=1}^{J}q_{j}\psi_{j})}{1-r}$$
(2)$$\mathrm{ARR}=\frac{\left(1-{\mathrm{PAF}}_{H}\right)(\sum_{j=1}^{J}q_{j}\psi_{j})}{r}$$
where PAF_H_ is the PAF associated with the risk factors in the high-risk group, *q_j_* denotes the proportion of deaths at exposure level j in the low-risk group, *ψ_j_* the rate ratio of risk factor at exposure category *j* relative to the referent category (*j* = 1 denotes the referent category) in the high-risk group, and r is the ratio of overall rate of high-risk group to the low-risk group. In this study, r was ratio of mortality rate between two groups, and the value was 1.86. Statistical analyses were conducted using SAS version 9.3 (SAS Institute, Inc., Cary, NC, USA), and figures were drawn using R 4.1.2 (R core team, R Foundation for Statistical Computing, Vienna, Austria). A *p* value < 0.05 was considered statistically significant.

## 3. Results

A total of 256 deaths were observed among 2446 SLE patients by 2021. Overall, a positive association was found between the initial SLEDAI scores on admission and mortality of SLE (*p* < 0.001), and it is shown from a linear pattern (Figure 1A). Associations between each item in the SLEDAI score and mortality among SLE patients are presented in Appendix A.

By the initial SLEDAI scores on admission, 7.7% (83) and 12.8% (176) of deaths were followed in the low–moderate and high activity groups, with a mortality rate (per 1000 person years) of 7.7 and 14.0, respectively (*p* < 0.001). The characteristics of the study patients with SLE for the two groups are shown in Table 1. The median (IQR) of the SLEDAI scores on admission for the low–moderate and high activity groups was 9.0 (6.0, 10.0) and 18.0 (15.0, 23.0), respectively (*p* < 0.001). Compared with the low–moderate group, the SLE patients in the high activity group had younger ages, lower survival time, higher SLEDAI scores at discharge, a higher proportion of organ (except ocular) involvement, higher abnormal laboratory tests on admission and higher medication use of glucocorticoids and immunosuppressives. Moreover, the Kaplan–Meier survival curves also revealed a significantly higher cumulative mortality in the high activity group than that in the low–moderate group (Figure 1B).

The major risk factors associated with the death of patients with SLE in the low–moderate and high activity groups are shown in Table 2. After adjusting for confounding factors, the risk factors positively associated with death were organ involvement, including neuropsychiatric, cardiopulmonary and gastrointestinal involvement, anemia, hypoalbuminemia, a decrease in the creatinine clearance rate, and antimalarial drug nonuse in both groups, whereas having an SLEDAI score ≥ 7 at discharge and comorbidities were only significant in the low–moderate activity group and having mucocutaneous involvement was only significant in the high activity group.

The individual PAF, RAR and ARR of the major risk factors in the low–moderate and high activity groups are presented in Figure 2. The most important contributor to death for each group was anemia, with a PAF (95% CI) of 40.4 (14.9, 61.8) in the high activity group and 37.5 (11.9, 57.6) in the low–moderate group, respectively. Hypoalbuminemia and a decrease in the creatinine clearance rate accounted for 20–30% of deaths for each group. Between the two groups, anemia (RAR = 23.2%, ARR = 1.66) was the most important factor contributing to the mortality differences, which means 23.2% of total deaths in the high activity group would be prevented if it had the same distribution of risk factors as the low–moderate group, and the rate ratio of the high activity group to the low–moderate group would decline from 1.86 to 1.66. In addition, cardiopulmonary involvement, hypoalbuminemia, neuropsychiatric involvement and decrease in the creatinine clearance rate could explain 10–20% of the mortality gradient between the two different risk groups. In particularly, antimalarial drug nonuse accounted for more than 35% of deaths in each group but explained 3.6% of the mortality differences between the two groups.

We further selected risk factors with individual RARs above 10% or PAFs above 20% to estimate their joint effects, and the combination of PAF, RAR and ARR are shown in Table 3. Cardiopulmonary involvement combined with either anemia, neuropsychiatric involvement or hypoalbuminemia accounted for 33.4 and 54.2% of total deaths in the two groups, which was higher than the sum of the two individual factors, suggesting potential interactions between those factors. Moreover, cardiopulmonary involvement combined with anemia had the highest RAR of 39.8%, with an ARR of 1.52. The joint PAFs of every other two factors working together were higher than the individual PAF of each risk factor, and the combination of more than two factors was not calculated because of the limited numbers in the cells after cross-grouping.

## 4. Discussion

To our knowledge, the present large multicenter retrospective study is the first to quantify the PAF of the risk factors of mortality and mortality differences due to variations in the distribution of risk factors between low–moderate and high disease activity in SLE patients. We found up to 40.4% of deaths could be attributable to certain risk factors, with anemia, antimalarial drug nonuse and hypoalbuminemia being the leading contributors. Moreover, up to 23.2% of death differences between the low–moderate and high activity groups could be explained by the distribution of individual risk factors, with the top three being anemia, cardiopulmonary involvement and hypoalbuminemia.

Compared with previous well-recognized findings, the results of the present study were validated as follows. First, consistent with the finding in previous studies [22,23,24], this study confirmed the SLEDAI score was positively associated with the mortality of SLE but also first observed the association was in a linear dose–response relationship. In addition, the magnitude of associations for each item in the SLEDAI with SLE death was related to their weight, with the highest HR in the item with a weighting of 8, especially for cerebrovascular accident, visual disturbance and seizure. Finally, it was not an unexpected finding that the SLE mortality rate was nearly twice higher in the high activity group than the low–moderate group. The SLE mortality rate observed in this study was similar to the previous reported rate of 5.18 per 1000 person years in Asian SLE patients but was lower than that of Native American, whites and blacks with more than 20 per 1000 person years [25]. It will be of great interest in future studies to reveal the proportion of death excess attributed to the distribution of risk factors among SLE patients of different races and ethnicities.

Anemia could be due to many etiologies such as chronic disease, renal involvements, drugs and infection, and even be associated with disease activity [26,27]. We observed anemia accounted for the highest percentage of SLE deaths among the risk factors in both the low–moderate and high activity groups, even when 103 (4.2%) patients with hemolytic anemia were further excluded. There are a few studies focused on the effects of anemia on the mortality of SLE. Tseng et al. reported patients with low hemoglobin were prone to death in a retrospective study with 3218 SLE patients [28]; while Ahn SS et al. found a higher hemoglobin level was independently associated with a lower risk of mortality among 171 patients with lupus nephritis [29]. The association of low hemoglobin with higher mortality in SLE patients may be indirectly explained by anemia of inflammation, which is the most frequent anemic entity observed in chronic systemic inflammatory disorders, including SLE. In addition, anemia in SLE usually develops in the context of systemic inflammation due to the decreased production of erythrocytes and a reduction in erythrocyte survival [30]. Thus, although anemia is not a specific cause of death, the findings suggested that it may be used as a marker for predicting the mortality of SLE.

SLE can affect multiple organs, ranging from mild mucocutaneous involvement to severe involvements in renal, neuropsychiatric, cardiovascular and pulmonary systems and leads to high disease activity and death [31,32]. Most manifestations in those organ involvement are SLEDAI-related factors. Cardiopulmonary involvement, mainly represented by serositis, accounted for the highest percentage of deaths difference (18.3%) between the low–moderate and high activity groups in this study. Consistently, cardiovascular complications are increasingly considered as critical for the prognosis of SLE patients and are one of the leading causes of SLE death, especially in developed countries [33,34]. In addition, neuropsychiatric involvements only accounted for less than 10% of PAF because of their relatively low prevalence. However, the different prevalence of neuropsychiatric involvement alone or combined with cardiopulmonary involvement accounted for 13.6 and 30% of the death differences between the two risk groups, suggesting that a major reduction in deaths in the high activity group could be expected if it had a similar distribution as the low–moderate group regarding these two factors. Overall, remission or the lowest possible disease activity should be the main treatment target in the treat-to-target strategy for SLE to avoid organ damage and mortality [35], but special measures need to be focused on the high RAR of organ involvement in refractory patients with high disease activity. In particular, various treatment strategies, such as anti-inflammatory therapy, lipid-lowering therapy and antiplatelet agents, should be taken to reduce deaths from cardiopulmonary involvement in SLE patients [36].

With respective to the effect of medication use on the mortality of SLE, we found antimalarial drug nonuse was an important risk factor. Previously, we have reported that antimalarial drugs were associated with lower risk of mortality in SLE patients [13]. This study further suggested more than 35% of total deaths could be attributable to antimalarial drug nonuse and almost the same distribution of nonuse between low–moderate and high activity groups, making a very small RAR. Consistent with the previous analysis, we only found cyclophosphamide nonusers had higher risk of mortality among SLE patients with the high activity group [37], but the PAF was low and not significant. Due to the limited sample size, we combined other immunosuppressives as a whole and found no significant differences between the low–moderate and high activity groups, and no association with SLE mortality in each group.

There are some limitations in this study which should be addressed. First, the population of this study was hospitalized SLE patients, so the measurements of PAF and RAR are only applicable to hospitalized SLE patients, and the values of RAR will be larger when the milder patients are included in the low–moderate group. Second, some items in the SLEDAI score were subjective, so the information bias, such as recall bias, from patients and subjective judgements from doctors may exist in this study, which would cause the non-differential misclassification of exposures between the low–moderate and high activity groups, making the measurements of the differences conservative. Third, although it was designed as a longitudinal study to observe the dynamic change in indicators in predicting SLE mortality, only about 23.9% of patients could have dynamic laboratory tests as 50.6% (131) of deaths were observed in the first 5 five years in this study, and first-hospitalized data were also important in presenting markers in the early stages for early treatment, especially when it is hard to obtain the dynamic data in routine medical work. Fourth, the main renal involvement was proteinuria (71.7%) and a decrease in the creatinine clearance rate (52.2%), but only 74 (4.7%) SLE patients had a renal biopsy. Despite these limitations, the strength of our study is as a larger multicenter cohort study, including nearly 2500 initial hospitalized SLE patients and more than 250 deaths in long- term follow-up, allowing for multivariate analyses to estimate the PAF and RAR in different risk groups using SLEDAI scores.

## 5. Conclusions

The differences in the distribution of risk factors, including anemia, cardiopulmonary and neuropsychiatric involvements and hypoalbuminemia contribute large differences in the mortality of SLE patients with different disease activity. Together with targeting remission and reducing disease activity, those risk factors should be used as early markers to improve the survival of SLE patients.

## Figures and Tables

**Figure 1 jcm-12-01061-f001:**
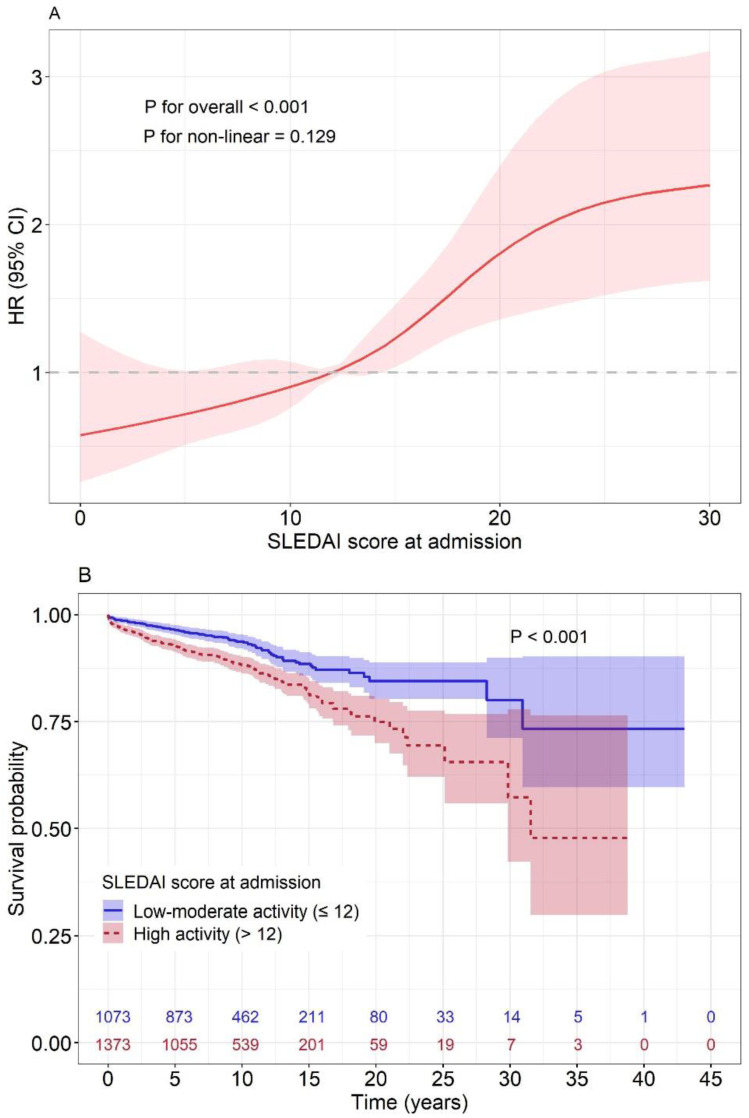
Association between SLE mortality and SLEDAI scores at admission of overall (**A**) and two groups of low–moderate activity and high activity (**B**). (**A**) The association between SLEDAI scores at admission and mortality in SLE using restricted cubic spline function with five knots, located at the 5th, 25th, 50th, 75th and 95th percentiles. The HRs and 95% CIs were calculated using the score of 12 as reference value and were adjusted for sex (male = 1, female = 0), age (continuous), comorbidities (yes = 1, no = 0), glucocorticoids treatment (yes = 1, no = 0) and immunosuppressive treatment (yes = 1, no = 0). (**B**) The comparison of survival probability was performed between low–moderate activity and high activity groups using SLEDAI at admission using Kaplan–Meier survival curves and log-rank tests.

**Figure 2 jcm-12-01061-f002:**
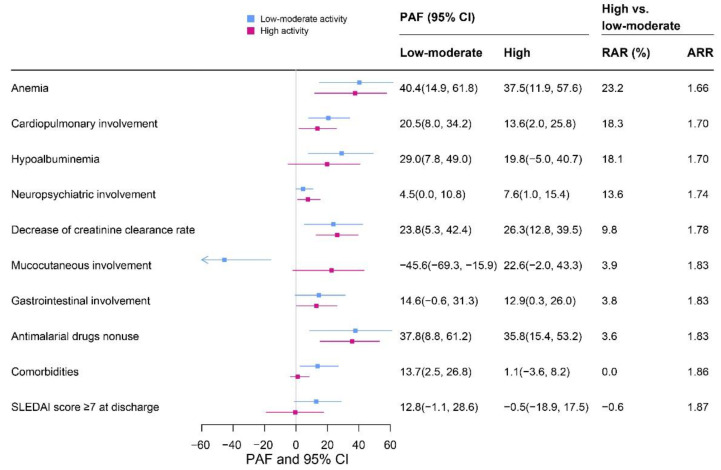
The individual population attributable fraction (PAF), relative attributable risk (RAR) and adjusted rate ratio (ARR) of risk factors in low–moderate activity and high activity groups in patients with SLE. The PAF was adjusted for sex (male = 1, female = 0), age (continuous), SLEDAI score on admission (continuous), comorbidities (yes = 1, no = 0, except for comorbidities), glucocorticoids treatment (yes = 1, no = 0) and immunosuppressive treatment (yes = 1, no = 0).

**Table 1 jcm-12-01061-t001:** Characteristics of study patients with SLE by initial SLEDAI score on admission.

	SLEDAI Score on Admission		
Variables	Low–Moderate Activity (≤12, *n* = 1073)	High Activity (>12, *n* = 1373)	*p*
Death	83(7.7)	176(12.8)	<0.001
Mortality rate (per 1000 person years, 95% CI)	7.6 (6.0, 9.4)	14.0 (1.2, 1.6)	<0.001
Age (years), median (IQR)	32.4 (24.2, 41.1)	31.5 (23.3, 40.0)	0.033
Diagnosis time (months), median (IQR)	91.0 (30.0, 377.0)	91.0 (38.0, 366.0)	0.275
Survival time (years), median (IQR)	9.0 (6.0, 13.5)	8.3 (5.3, 12.7)	<0.001
SLEDAI score on admission, median (IQR)	9.0 (6.0, 10.0)	18.0 (15.0, 23.0)	<0.001
SLEDAI score at discharge, median (IQR)	2.0 (0.0, 6.0)	6.0 (2.0, 13.0)	<0.001
Gender (female)	997 (92.9)	1267 (92.3)	0.551
Comorbidities	129 (12.0)	122 (8.9)	0.011
Organ involvement			
Mucocutaneous	640 (59.6)	949 (69.1)	<0.001
Neuropsychiatric	22 (2.1)	129 (9.4)	<0.001
Musculoskeletal	426 (39.7)	909 (66.2)	<0.001
Cardiopulmonary	137 (12.8)	386 (28.1)	<0.001
Gastrointestinal	334 (31.1)	502 (36.6)	0.005
Ocular	2 (0.2)	9 (0.7)	0.085
Renal	584 (54.4)	1007 (73.3)	<0.001
Hematological	431 (40.2)	719 (52.4)	<0.001
Laboratory tests on admission			
Anemia	560 (52.2)	983 (71.6)	<0.001
Abnormal white blood cell count	528 (49.2)	776 (56.5)	<0.001
Hypoalbuminemia	428 (39.9)	883 (64.3)	<0.001
Decrease in creatinine clearance rate	336 (31.3)	494 (36.0)	0.016
Anti-dsDNA positive	473 (44.1)	845 (61.5)	<0.001
Antinuclear antibody positive	896 (83.5)	1232 (89.7)	<0.001
Anti-Sm positive	277 (25.8)	451 (32.8)	<0.001
Abnormal C3 complement	538 (50.1)	1028 (74.9)	<0.001
Abnormal C4 complement	387 (36.1)	773 (56.3)	<0.001
Medication use			
Glucocorticoids	960 (89.5)	1310 (95.4)	<0.001
Immunosuppressive	746 (69.5)	1013 (73.8)	0.020
Antimalarial drugs	444 (41.4)	544 (39.6)	0.379
Cyclophosphamide	429 (40.0)	648 (47.2)	<0.001
Other immunosuppressives	293 (27.3)	339 (24.7)	0.142

Data are presented as the number (%) unless otherwise stated; SLEDAI: SLE disease activity index 2000.

**Table 2 jcm-12-01061-t002:** Major risk factors associated with death in patients with SLE by initial SLEDAI score on admission.

	Low–Moderate Activity (≤12)			High Activity (> 12)		
Variables	Survival No. (%)	Death No. (%)	Adjusted HR (95% CI) ^a^	Survival No. (%)	Death No. (%)	Adjusted HR (95% CI) ^a^
SLEDAI score at discharge						
0–6	796 (80.4)	57 (68.7)	1.00	619 (51.7)	85 (48.3)	1.00
≥7	194 (19.6)	26 (31.3)	1.69 (1.02, 2.82)	578 (48.3)	91 (51.7)	0.99 (0.73, 1.35)
Comorbidities						
No	882 (89.1)	62 (74.7)	1.00	1096 (91.6)	155 (88.1)	1.00
Yes	108 (10.9)	21 (25.3)	2.18 (1.30, 3.64)	101 (8.4)	21 (11.9)	1.11 (0.69, 1.78)
Mucocutaneous involvement						
No	380 (38.4)	53 (63.9)	1.00	379 (31.7)	45 (25.6)	1.00
Yes	610 (61.6)	30 (36.1)	0.44 (0.28, 0.70)	818 (68.3)	131 (74.4)	1.44 (1.02, 2.02)
Neuropsychiatric involvement						
No	973 (98.3)	78 (94.0)	1.00	1098 (91.7)	146 (83.0)	1.00
Yes	17 (1.7)	5 (6.0)	3.86 (1.55, 9.63)	99 (8.3)	30 (17.0)	1.80 (1.17, 2.76)
Cardiopulmonary involvement						
No	879 (88.8)	57 (68.7)	1.00	880 (73.5)	107 (60.8)	1.00
Yes	111 (11.2)	26 (31.3)	2.90 (1.79, 4.68)	317 (26.5)	69 (39.2)	1.53 (1.12, 2.10)
Gastrointestinal involvement						
No	687 (69.4)	52 (62.7)	1.00	772 (64.5)	99 (56.3)	1.00
Yes	303 (30.6)	31 (37.3)	1.64 (1.04, 2.57)	425 (35.5)	77 (43.8)	1.42 (1.05, 1.91)
Anemia						
No	489 (49.4)	24 (28.9)	1.00	358 (29.9)	32 (18.2)	1.00
Yes	501 (50.6)	59 (71.1)	2.31 (1.43, 3.75)	839 (70.1)	144 (81.8)	1.85 (1.26, 2.72)
Hypoalbuminemia						
No	610 (61.6)	35 (42.2)	1.00	443 (37.0)	47 (26.7)	1.00
Yes	380 (38.4)	48 (57.8)	2.01 (1.29, 3.13)	754 (63.0)	129 (73.3)	1.37 (0.98, 1.92)
Decrease in creatinine clearance rate						
No	695 (70.2)	42 (50.6)	1.00	796 (66.5)	83 (47.2)	1.00
Yes	295 (29.8)	41 (49.4)	1.93 (1.24, 3.00)	401 (33.5)	93 (52.8)	1.99 (1.47, 2.69)
Antimalarial drug nonuse						
No	422 (42.6)	22 (26.5)	1.00	499 (41.7)	45 (25.6)	1.00
Yes	568 (57.4)	61 (73.5)	2.06 (1.25, 3.39)	698 (58.3)	131 (74.4)	1.93 (1.36, 2.72)

^a^ Adjusted for sex (male = 1, female = 0), age (continuous), SLEDAI score on admission (continuous), comorbidities (yes = 1, no = 0), glucocorticoids treatment (yes = 1, no = 0) and immunosuppressive treatment (yes = 1, no = 0), and the above adjusted variables would be excluded when it was the analytical variable. SLEDAI: SLE disease activity index 2000.

**Table 3 jcm-12-01061-t003:** The population attributable fraction (PAF), relative attributable risk (RAR) and adjusted rate ratio (ARR) for the combination of risk factors in low–moderate activity and high-activity of patients with SLE.

Variables	Low–Moderate Activity (≤12)		High Activity (>12)		High vs. Low–Moderate	
	Case (%)	PAF% (95% CI) ^a^	Case (%)	PAF% (95% CI) ^a^	RAR (%)	ARR
Anemia						
Cardiopulmonary involvement	71.1	39.4 (13.5, 61.1)	82.4	37.7 (11.6, 58.0)	39.8	1.52
Hypoalbuminemia	79.5	55.1 (30.0, 74.6)	90.3	59.3 (33.1, 76.9)	31.7	1.59
Neuropsychiatric involvement	84.3	54.2 (21.8, 76.5)	92.0	51.0 (14.4, 73.6)	35.5	1.55
Decrease in creatinine clearance rate	69.9	39.5 (14.6, 60.7)	85.2	45.0 (18.7, 64.6)	25.5	1.64
Antimalarial drug nonuse	79.5	45.2 (14.0, 68.6)	86.9	37.7 (5.1, 61.0)	26.0	1.64
Cardiopulmonary involvement						
Hypoalbuminemia	71.1	48.8 (27.2, 67.4)	77.3	36.9 (14.9, 55.1)	38.6	1.53
Neuropsychiatric involvement	75.9	43.5 (15.0, 66.0)	85.8	33.4 (0.3, 57.4)	30.5	1.60
Decrease in creatinine clearance rate	60.2	36.3 (16.2, 55.1)	71.0	31.2 (11.4, 48.5)	26.1	1.64
Antimalarial drug nonuse	69.9	34.7 (7.9, 57.4)	75.0	25.3 (1.1, 45.6)	21.4	1.68
Hypoalbuminemia						
Neuropsychiatric involvement	71.1	45.9 (22.9, 65.5)	84.7	44.5 (18.7, 64.0)	33.2	1.57
Decrease in creatinine clearance rate	36.1	25.2 (11.8, 39.4)	48.3	19.7 (6.0, 33.3)	21.4	1.68
Antimalarial drug nonuse	65.1	41.0 (20.0, 60.0)	70.5	35.4 (16.9, 51.6)	21.6	1.67
Neuropsychiatric involvement						
Decrease in creatinine clearance rate	60.2	32.0 (10.5, 52.0)	78.4	27.5 (0.8, 48.9)	21.5	1.68
Antimalarial drug nonuse	77.1	45.0 (16.6, 67.3)	82.4	22.6 (−10.5, 47.8)	16.5	1.72
Decrease in creatinine clearance rate						
Antimalarial drug nonuse	51.8	26.3 (7.3, 45.1)	59.7	30.9 (16.1, 44.9)	9.5	1.78

^a^ Adjusted for sex (male = 1, female = 0), age (continuous), SLEDAI score on admission (continuous), comorbidities (yes = 1, no = 0), glucocorticoids treatment (yes = 1, no = 0) and immunosuppressive treatment (yes = 1, no = 0).

## Data Availability

The datasets used and analyzed in this study are available from the corresponding author upon reasonable request.

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
