# Peer review of "Comparison of Contributors to Mortality Differences in SLE Patients with Different Initial Disease Activity: A Larger Multicenter Cohort Study"

_jcm, 2023, doi:10.3390/jcm12031061_

Round 1

Reviewer 1 Report

It would be interesting to make a comparison with respect to the group of patients with low activity. Separating with older or younger SLEDAI generates very heterogeneous groups.

Assuming. the impact on mortality that cardiovascular involvement has, cardiovascular risk factors are not defined or categorized.

The impact of biologic drugs is also not mentioned.

Author Response

Reviewer 1

Comments and Suggestions for Authors

It would be interesting to make a comparison with respect to the group of patients with low activity. Separating with older or younger SLEDAI generates very heterogeneous groups.

Response: Thank you very much for the comments and suggestions.

Assuming. the impact on mortality that cardiovascular involvement has, cardiovascular risk factors are not defined or categorized.

Response: Thank you for your kind suggestion. We collected clinical manifestations but not its risk factors from SLE patients, and cardiovascular involvement was not easy to be observed at the early stages of the disease. We mainly observed serositis, interstitial lung disease, myocarditis, and pulmonary arterial hypertension, and a few others. Thus, cardiopulmonary involvement was analyzed this study. The following sentence has been added in the first paragraph on page 3 of the revised manuscript:

“Cardiopulmonary involvement mainly included serositis, interstitial lung disease, myocarditis, and pulmonary arterial hypertension, and a few other manifestations such as pulmonary hemorrhage/vasculitis, valvular dysfunction, arrhythmia and cardiac failure.”

The impact of biologic drugs is also not mentioned.

Response: Thank you for raising this point. Medical records from first-hospitalized SLE patients during 1999-2009 were collected, and only 3 patients used rituximab and 2 patients used infliximab. According to your suggestion, the following sentence has been added in end of the first paragraph on page 3 of the revised manuscript:

“Biologic drugs such as rituximab and infliximab were also collected, but was not analyzed due to its very small sample size.”

Reviewer 2 Report

It is an interesting clinical study to explore the etiology of risk factors and quantify the mortality differences in systemic lupus erythematosus (SLE) with different initial disease activity by using Jiangsu Lupus database with medical records from first-hospitalized patients among 26 medical centers in China during 1999 to 2009. The authors used population attributable fraction, relative attributable risk and adjusted relative risk  to quantify initial SLEDAI score differences in mortality attributable to risk factors. Finally, they concluded that differences in the distribution of risk factors including anemia, cardiopulmonary and neuropsychiatric involvements, and hypoalbuminemia contribute large differences in the mortality of SLE patients with different disease activity.

The manuscript is well written in English and relevant to the clinical application. There were only minor issues needed to be clarified as follows.

1.     In Table 1, Medication use included Glucocorticoids, Immunosuppressive, Antimalarial drugs, Cyclophosphamide and Other Immunosuppressives. The authors should clarify the difference between “Immunosuppressive” and “Other Immunosuppressives”.

2.    In the Study design and patients of Methods section, the authors should cite appropriate reference 16 for the SLE disease activity index 2000 score, i.e., Gladman DD, Ibañez D, Urowitz MB. Systemic lupus erythematosus disease activity index 2000. J Rheumatol 2002;29:288-91. In addition, they should define the “positive” anti-dsDNA, antinuclear antibody and anti-Sm by providing the laboratory titers.

3.   In the Abstract section, the first sentence could be revised as “To explore the etiology of risk factors and quantify the mortality differences in systemic lupus erythematosus (SLE) patients with different initial disease activity”.

Author Response

Reviewer 2

It is an interesting clinical study to explore the etiology of risk factors and quantify the mortality differences in systemic lupus erythematosus (SLE) with different initial disease activity by using Jiangsu Lupus database with medical records from first-hospitalized patients among 26 medical centers in China during 1999 to 2009. The authors used population attributable fraction, relative attributable risk and adjusted relative risk to quantify initial SLEDAI score differences in mortality attributable to risk factors. Finally, they concluded that differences in the distribution of risk factors including anemia, cardiopulmonary and neuropsychiatric involvements, and hypoalbuminemia contribute large differences in the mortality of SLE patients with different disease activity.

Response: Thank you very much for your kind comments.

The manuscript is well written in English and relevant to the clinical application. There were only minor issues needed to be clarified as follows.

  1. In Table 1, Medication use included Glucocorticoids, Immunosuppressive, Antimalarial drugs, Cyclophosphamide and Other Immunosuppressives. The authors should clarify the difference between “Immunosuppressive” and “Other Immunosuppressives”.

Response: We appreciate your comments on this. The following sentences have been added in end of the first paragraph on page 3 of the revised manuscript:

“Immunosuppressive drugs included antimalarial drugs, cyclophosphamide, leflunomide, azathioprine, methotrexate, mycophenolate mofetil, tripterygium, ciclosporin, and tacrolimus. According to the proportion, we analyzed antimalarial drugs and cyclophosphamide separately, and combined others as other immunosuppressives.”

  1. In the Study design and patients of Methods section, the authors should cite appropriate reference 16 for the SLE disease activity index 2000 score, i.e., Gladman DD, Ibañez D, Urowitz MB. Systemic lupus erythematosus disease activity index 2000. J Rheumatol 2002;29:288-91. In addition, they should define the “positive” anti-dsDNA, antinuclear antibody and anti-Sm by providing the laboratory titers.

Response: Thank you for raising this point. We have revised the reference 16 according your comments. The positive anti-dsDNA was detected by immunoblotting, or value >100 by enzyme immunoassays, the positive antinuclear antibody was detected by value > 1:40 in immumofluorescence method, and the positive anti-Sm positive was detected by immunoblotting. The following part of sentence has been revised in the first paragraph on page 3 of the revised manuscript:

“anti-dsDNA positive (immunoblotting, or >100 by enzyme immunoassays), antinuclear antibody positive (> 1:40 by immumofluorescence method), anti-Sm positive (immunoblotting)”

  1.  In the Abstract section, the first sentence could be revised as “To explore the etiology of risk factors and quantify the mortality differences in systemic lupus erythematosus (SLE) patients with different initial disease activity”.

Response: Thank you for your suggestion. We have revised the first sentence of the Abstract according your suggestion.